# Time Decomposition and Short-Term Forecasting of Hydrometeorological Conditions in the South Baltic Coastal Zone of Poland

**Jacek Tylkowski [1],\* and Marcin Hojan [2]**

[1] Institute of Geoecology and Geoinformation, Faculty of Geographical and Geological Sciences, Adam Mickiewicz University, Krygowski 10, 61-680 Poznań, Poland

[2] Institute of Geography, Department of Landscape History Research, Kazimierz Wielki University, Kościeleckich Square 8, 85-033 Bydgoszcz, Poland; homar@ukw.edu.pl

\* Correspondence: jatyl@amu.edu.pl

**Abstract:** This article presents an analysis of time-series for hydrometeorological conditions determining the behavior of the natural environment in the South Baltic coastal zone of Poland. The analysis is based on monthly data for average air temperature, total atmospheric precipitation, and average sea level during the 50-year period from 1966–2015 for three coastal stations in Hel, Ustka, and Świnoujście. Time decomposition of these hydrometeorological conditions and formulation of short-term forecasts were carried out using ARIMA modelling. This study identifies the seasonal and non-seasonal parameters that determine both current and future hydrometeorological conditions. Moreover, it indicates the spatial differences among features of the analyzed time-series, estimated parameters of the selected models, and forecasts. The ARIMA models used for the Polish Baltic coastal zone are somewhat spatially homogenous. This is especially true of the models for average monthly air temperature, which are identical across the entire coastal zone $(2,0,1)(2,1,0)_{12}$. Very similar are the models for average monthly sea level across the central and west coast $(1,0,0)(1,1,0)_{12}$. The model for the east coast, however, was determined to be slightly different $(2,0,2)(2,1,0)_{12}$. In contrast to those for air temperature and sea level, the models used for atmospheric precipitation were different for each site. Among the parameters modelled, the effect of $AR(p)$ processes was greater than that of $MA(q)$ processes. The monthly models for Ustka are an example of this: average air temperature $(2,0,1)(2,1,0)_{12}$, atmospheric precipitation $(0,0,3)(2,1,0)_{12}$, and average sea level $(1,0,0)(1,1,0)_{12}$. Time decomposition of extreme hydrometeorological conditions has an important utilitarian significance. The climate of the Polish Baltic coastal zone is getting warmer, the sea level is rising, and the frequency of extreme hydrometeorological events is increasing. Time decomposition of hydrometeorological conditions based on monthly data did not reveal long-term trends. In the last half-century, hydrometeorological conditions have been conducive to erosion of coastal dunes and cliffs. These factors determine changes in the natural environment and limit the development potential of the coastal zone. The time decomposition, modelling, and forecasting of hydrometeorological conditions are thus very important for many areas of human activity, especially those related to management, protection, and development of the coast.

**Keywords:** ARIMA model; time decomposition; hydrometeorological conditions; South Baltic coast

## 1. Introduction

The decomposition of time-series is an important task in research. Time-series data varies according to season and are widely used in forecasting and analysis of natural environment cycles. To determine the behavior of contemporary geo-ecosystems of the coastal zone, we must examine

the properties of time-series for meteorological conditions (e.g., atmospheric precipitation and air temperature) and hydrological conditions (e.g., sea level). Hydrometeorological conditions determine the potential initiation, intensity, and duration of natural processes that affect the behavior of biotic and abiotic elements in the natural environment.

Many methods of forecasting time-series for hydrometeorological data are based on analysis of historical data. The purpose of time-series analysis is to build models that accurately describe the temporal dynamics of hydrometeorological conditions and enable forecasting of unknown values. Further observation of time-series usually reveals significant, non-random relationships. The main task of time-series analysis is to identify the nature and strength of these relationships for the prediction of future values [1–3].

Classic statistical models like autoregressive integrated moving-average(ARIMA) are most often used in the analysis and forecasting of one-dimensional time-series. An advantage of these models is their extensive mathematical aspect, which allows effective identification of their parameters, as well as comprehensive assessment of their appropriateness [4–6].

In recent years, one of the most popular ways of time-series modelling is ARIMA modelling. Its main aim is to carefully and rigorously study the past observations of a time-series to develop an appropriate model which can predict future values for the series. It has three control constants (trend, seasonal, and irregular influence), which can control and manage influence of time segmentation through the specific time duration [7]. The ARIMA models are now widely used for various applications, such as natural environment [8–11], medicine [12], and engineering [13,14]. Autoregressive integrated moving average modelling is often used in many areas of the economy, e.g., in forecasting tourist traffic at airports [15], in road transport [16,17], in real estate price analysis [18], unemployment analysis [19], in studies of electricity consumption and their price forecasts [20,21], and in wind energy and runoff forecasts [22,23]. Short-term forecasts using the ARIMA model are also used in fisheries [24] and for the needs of forestry, such as the prediction of drought, risk of fires, and prognosis of tree diseases [25]. Autoregressive integrated moving average modelling is also used in studying sea coastal zones, e.g., for forecasts of the coast line and sea level changes [26–28] and in zooplankton studies [29]. Other models are also used to analyze meteorological conditions, e.g., Copula-GARCH for droughts [30] or a hybrid stochastic rainfall model for hourly rainfall in a yearly time scale [31].

The main aim of this work is therefore time decomposition and forecasting of hydrometeorological conditions in the Polish Baltic coastal zone, on the basis of data collected from three measuring stations in Świnoujście (west, Szczecin coastal region), Ustka (central, Koszalin coastal region), and Hel (east, Gdańsk coastal region). Time decomposition using ARIMA non-seasonal modelling and SARIMA seasonal modelling allows determination of the specificity of hydrometeorological conditions during the 50-year period from 1966–2015, as well as the formulation of a prognosis through 2020.

The specific objectives of the work are as follows:

- Identification of time-series characteristics (identifying the trend and seasonality of hydrometeorological data, determining the need for data transformation or data differentiation, application of diagnostic tests confirming the stationarity of the time-series, selection of additive or multiplicative time decomposition model).
- Estimation of parameters and diagnostic testing of models (selection of models including non-seasonal [ARIMA(p,d,q)] and seasonal [SARIMA(p,d,q)(P,D,Q)s] factors based on goodness-of-fit criteria (Akaike Information Criterion (AIC), Corrected Akaike Information Criterion (AICC), Bayesian Information Criterion (BIC)), prediction-error criteria (mean absolute error (MAE), root mean squared error (RMSE), mean absolute percentage error (MAPE), mean absolute scaled error (MASE), $V\epsilon$,) and verification of residuals normality by the Shapiro–Wilk test).
- Model implementation and forecasting (out-of-sample accuracy analysis and short-term forecast for prediction intervals containing confidence levels of 80% and 95%).

## 2. Materials and Methods

### 2.1. Study Area

The Polish coastline is nearly 500 km long and represents two basic types of coastline: dunes and cliffs. Eighty percent of the coastline is made up of dunes that have developed within the Holocene, most often in the form of sandy barriers 2–35 m high. Around 15% of the coastline is made up of cliffs under 100 m high, formed during the Pleistocene Epoch. The remainder is made up of low coasts (only a few meters above sea level) in the estuary sections of rivers and near floodplains or organic accumulations.

Time decomposition and modelling were performed using monthly hydrometeorological data from 1966–2015 on average air temperature, total atmospheric precipitation, and average sea level. The data, taken from three costal stations in Świnoujście, Ustka, and Hel (Figure 1), was provided by the Institute of Meteorology and Water Management in Warsaw (https://danepubliczne.imgw.pl).

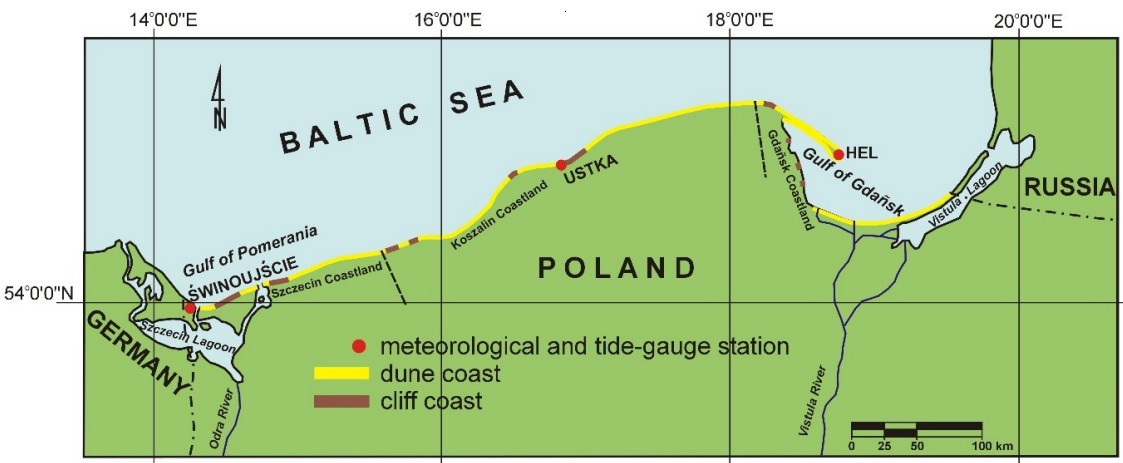

**Figure 1.** Area of research and location of measurement stations in the Polish Baltic coastal zone.

### 2.2. Methodology

This article presents the time decomposition and forecasting of one-dimensional time-series of hydrometeorological conditions using ARIMA modelling. Autoregressive integrated moving average (*p*,*d*,*q*) modelling is one of the most effective methods of time-series forecasting [32,33]. The ARIMA models account for integrated autoregressive and moving-average processes. They can be used to model stationary time-series or non-stationary time-series that have been adjusted for stationarity. There are three basic types of models: autoregressive models (AR); moving-average models (MA); and mixed autoregressive and moving-average models (ARMA). The "I" symbol in ARIMA indicates that the time-series has undergone a differencing operation. Also used in this work were seasonal ARIMA models, i.e. SARIMA (*P*,*D*,*Q*)$_s$, which allow modelling of seasonal data. A SARIMA model is essentially an ARIMA model that includes seasonal terms [34]. It is a more effective model than classical decomposition, as it allows dynamic (random) changes in the periodic component. After the transformation of data using differencing operations, a stationary ARMA model was adjusted to both the ARIMA and SARIMA models. It was assumed that SARIMA is a more effective model than classical decomposition. Yet ARIMA and SARIMA models are often not distinguished [3]. In such cases, ARIMA refers to the most general (i.e., seasonal) variant (ARIMA ≡ SARIMA).

The seasonal ARIMA (*p*,*d*,*q*) (*P*,*D*,*Q*)$_m$ process, also referred to as SARIMA (*p*,*d*,*q*) (*P*,*D*,*Q*)$_m$ is given by [7,35]:

$$\Phi(B^m)\phi(B)(1 - B^m)^D(1 - B)^d Y_t = c + \Theta(B^m)\theta(B)\varepsilon_t \tag{1}$$

where: $m$ is the seasonal period; $Y_t$ is a stationary stochastic process; $c$ is the constant; $\varepsilon_t$ is the error or white noise disturbance term; $B$ is a backward operator; and $\Phi(z)$ and $\Theta(z)$ are polynomials of orders $P$ and $Q$, respectively, each containing no roots inside the unit circle. If $c \neq 0$, there is an implied polynomial of order $d + D$ in the forecast function.

Choosing the appropriate ARIMA models [3,36,37] for air temperature, atmospheric precipitation, and sea level involved identification of time-series characteristics, estimation of parameters, diagnostic testing of the models, and implementation of the models for short-term forecasting.

### 2.2.1. Identification of Time-Series Characteristics

To stabilize variance, along with the level of the time-series, it was not necessary to transform the data, e.g., via Box–Cox transformation. This was because no outlier values, heterogeneous data variability, or amplitude of seasonal fluctuations in subsequent periods were found. The autocorrelation function (ACF) and partial autocorrelation function (PACF) were used to identify characteristics of time-series and determine their stationarity. The ACF is a natural generalization of the linear correlation coefficient and a measure of time-series observations distant by $h$ time units (*lag*). It allows distinction of stationary and non-stationary models, determination of seasonality, and identification of orders for the moving-average model MA($q$). PACF allows determination of direct relationships between observations distant by $h$ time units, and is particularly useful for identifying strong upwards data trends and autoregressive models AR($p$). For the monthly data analyzed, no positive correlations were found for *lag* = 1, which confirms the absence of long-term trends. However, for the hydrometeorological conditions analyzed, a seasonal relationship was found for values distant by *lag* = 12. Scatterplots for *lag* = 12 confirmed the seasonality of data, which is linked to the variability of hydrometeorological conditions over the course of the year. For this reason, one-time, interventional differencing and seasonal adjustment of data for *lag* = 12 were performed in order to stabilize the average. Order $D$ in the SARIMA model was assumed as 1 (multiplicity of differencing with *lag* = 12 to eliminate seasonality), and order $d$ in the ARIMA was assumed as 0 (multiplicity of differencing with *lag* = 1—no need to eliminate the trend). The course of the ACF function after differencing confirmed the stationarity of the time-series analyzed, in accordance with the principle that the ACF of stationary series is significantly greater than 0 in case of 5 lags at most for significance level $\alpha = 0.05$. Dickey–Fuller tests also revealed the stationarity of the time-series. Due to the constant amplitude of seasonal fluctuations, an additive model of time decomposition without the trend component was used. The formula was $X_t = s_t + Z_t$ ($X_t$—time-series, $s_t$—seasonal component, $Z_t$—random disruption).

### 2.2.2. Estimation of Parameters and Diagnostic Testing of Models

For stationary time-series, potential models were identified using the ACF and PACF functions. The auto.arima function then allowed determination of AR($p$,$P$) and MA($d$,$D$) parameters. The auto.arima function is based mainly on the Hyndman–Khandakar algorithm, which combines unit element tests, minimization of corrected Akaike tests (AICc), and the maximum likelihood method to obtain the model most appropriate for the data. The criteria of goodness-of-fit, based on the information criterion, were taken into account [38]:

Akaike Information Criterion (AIC)

$$\text{AIC}(p, q) = -2\ln(L) + 2(p + q + k + 1) \tag{2}$$

Corrected Akaike Information Criterion (AICC)

$$\text{AICC}(p, q) = -2\ln(L) + 2(p + q + k + 1)\, n/(n - p - q - k - 2) \tag{3}$$

Bayesian Information Criterion (BIC)

$$\text{BIC}(p, q) = -2\ln(L) + (p + q + k + 1)\ln n \tag{4}$$

where: *p*—autoregression parameter, *q*—moving average parameter, *L*—likelihood, *k*—number of model parameters, $k = 0$ (c = 0), $k = 1$ ($c \neq 0$), *n*—number of data, sample size.

Model diagnostics were based on analysis of the properties of model residual series using the Shapiro–Wilk test of normality. The *p*-values in the tests were greater than the established level of significance ($\alpha = 0.05$). The model residuals thus confirmed the randomness and lack of temporal correlation between observations, as there were no significant values of the ACF and PACF functions for the residuals. The white noise processes $WN(\sigma^2)$ obtained from the residuals assume no temporal correlation between hydrometeorological conditions [39]. These processes then enabled formulation of forecasts.

2.2.3. Implementation of Models for Short-Term Forecasts

The models were divided into a training set (for model matching and forecasting for January 1966–December 2003) and a test set (to assess the accuracy of the forecast for January 2004–December 2015) for further forecasting of hydrometeorological phenomena over the next four years (i.e., though 2020). Confidence levels of 80% and 95% were established for the prediction intervals. The final model was selected based on analysis of prediction-error criteria, namely the mean absolute error (MAE), root mean squared error (RMSE), mean absolute percentage error (MAPE), and mean absolute scaled error (MASE), where $X_i$ was the actual value, and $F_i$ was the forecasted value [3]:

$$\text{MAE} = \frac{1}{n}\sum_{i=1}^{n}|X_i - F_i| \quad \text{RMSE} = \frac{1}{n}\sum_{i=1}^{n}(X_i - F_i)^2 \tag{5}$$

$$\text{MAPE} = \frac{1}{n}\sum_{i=1}^{n}\left\{\frac{X_i - F_i}{X_i}100\%\right\} \quad \text{MASE} = \frac{1}{n}\sum_{i=1}^{n}|q_i| \tag{6}$$

Also used was the prediction accuracy index $V_\epsilon$, which determines what percentage of the average value of the explanatory variable is a standard deviation from the residual. The values obtained for $V_\epsilon$ were <3%, which confirms that the forecasts were well-matched.

## 3. Results

### 3.1. Meteorological Background

The average annual air temperature from 1966–2015 ranged from 8.6 °C in Świnoujście to 8.2 °C in Ustka and Hel. Precipitation conditions were more spatially various than thermal conditions. The highest average annual sum of atmospheric precipitation was 696.5 mm in the central coastal zone of Ustka. In the eastern Baltic coastal zone of Hel, the average annual sum of atmospheric precipitation was lower by over 100 mm (585.3 mm). The lowest average annual sum of atmospheric precipitation was 548.9 mm in the western coastal zone of Świnoujście. The spatial variation of average sea level during the period was insignificant – from 501 cm in Świnoujście to 504 cm in Ustka and Hel. Monthly data from this period do not indicate any statistically significant trends in changes of the hydrometeorological conditions studied. However, the seasonal distribution of monthly values for sea level and thermal and precipitation conditions is linked to properties of areas in the relatively warm transitional climate zone [40]. On the South Baltic coast, the seasons are marked by air temperature distribution, with the warmest months occurring in the summer season (~17.5 °C in July and August) and the coldest months occurring in the winter season ($\sim -0.1$ °C in January). Atmospheric precipitation occurs throughout the year, with increased intensity during summer (60–70 mm monthly from July to September) and decreased intensity during winter (30–40 mm monthly from February to April). The Polish Baltic coastal zone has a marine-type climate characterized by high variability of weather conditions, relatively low annual air temperature amplitude, and relatively insignificant atmospheric precipitation. Thermal conditions are therefore fairly balanced in the entire zone, in contrast to atmospheric precipitation, which is characterized by greater spatial variability. In terms of

sea level, relatively high values occur in the cold half-year, especially during the storm period, which usually lasts from November to January; while the lowest sea levels occur during the springtime, usually from March to May. Sea levels are presented in cm using only the Baltic High System (BHS) (based on the Kronstadt sea-gauge). The estimated difference between the Normal-Null (NN) and BHS-based systems is about 15 cm (the Kronstadt system is higher). Although Poland uses a high system based on the Kronstadt sea gauge, the registration and recording of sea levels is based on the NN reference system [41]. It should be emphasized that the above regularities in the monthly distributions of hydrometeorological conditions (Figure 2) can differ significantly among individual years in the 1966–2015 period, which is typical for the area.

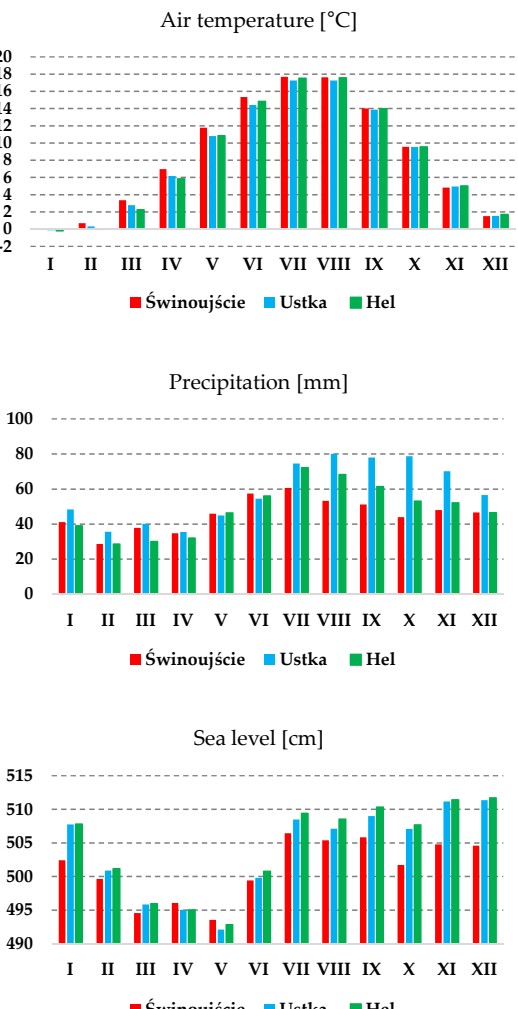

**Figure 2.** Average monthly values for air temperature, atmospheric precipitation, and sea level in Świnoujście, Ustka, and Hel from 1966–2015.

## 3.2. Identification of Time-Series Characteristics

Analysis of the time-series for monthly values of average sea level, average air temperature, and total atmospheric precipitation revealed very similar regularities in Hel, Ustka, and Świnoujście. Time decomposition of hydrometeorological conditions based on monthly data did not reveal long-term trends. However, some fluctuations caused by the change of seasons were noticeable. Additive decomposition of the time-series indicated the highest values for temperature and precipitation during the summer, that are associated with intense rainfall during storms. The lowest values of temperature and precipitation were during the winter. Sea level was significantly higher during the autumn and winter storm period, when there was frequent movement of barometric depression above Baltic

Sea. Seasonal fluctuations of hydrometeorological phenomena do not depend on the level of their time-series, as they were more or less constant. Due to the great similarity of the time decompositions of hydrometeorological conditions for the three measurement stations, the results for Ustka can be considered representative for the entire Polish Baltic coastal zone. They are presented in graphic form below (Figure 3).

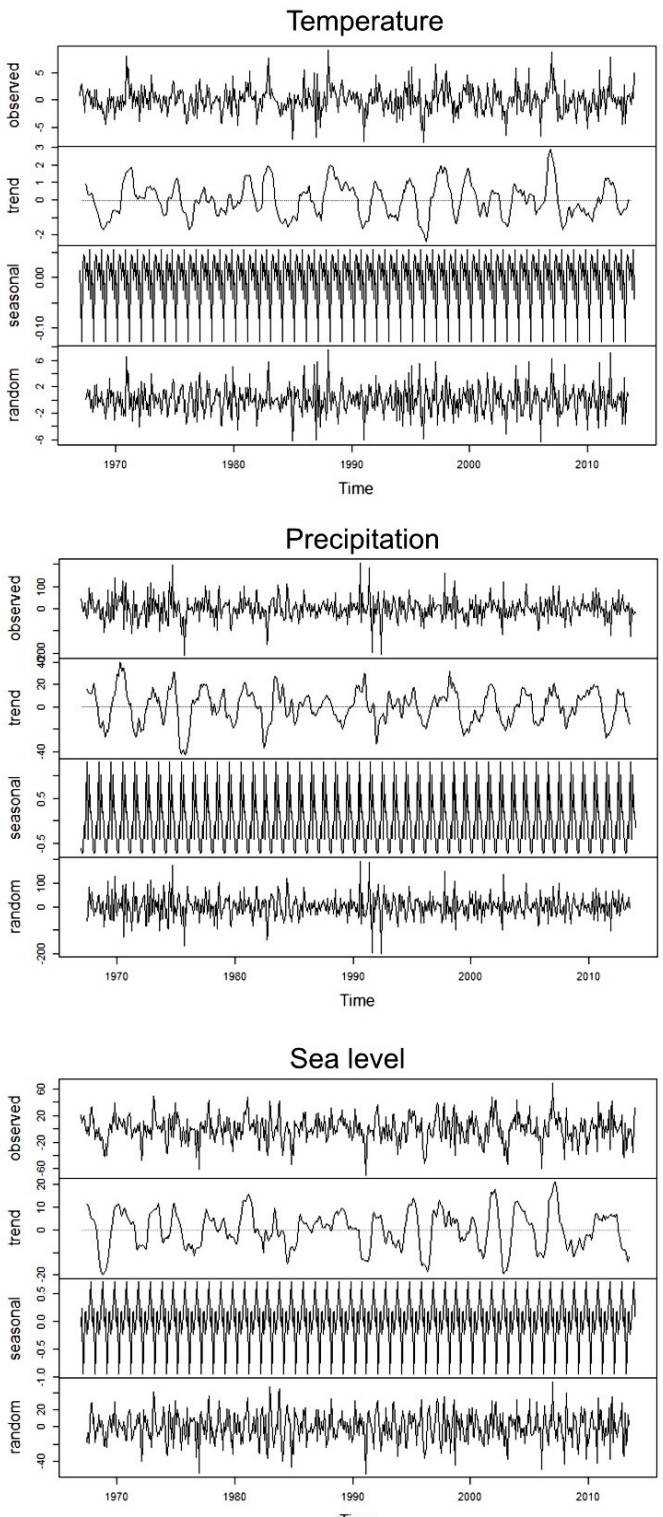

**Figure 3.** Additive decomposition for hydrometeorological conditions in Ustka (monthly intervals from 1966–2015).

An important issue in the analysis of time-series is temporal autocorrelation of data. Autocorrelation (ACF) and partial autocorrelation (PACF) of the hydrometeorological phenomena were conducted after differencing (Figure 4). For the ACF and PACF functions, one cycle was one year (12 months). The quick decay of the ACF and PACF functions, as well as the results of the Dickey–Fuller tests, confirmed the stationarity of the time-series and the absence of seasonality. Stationarity was essential for modelling and forecasting.

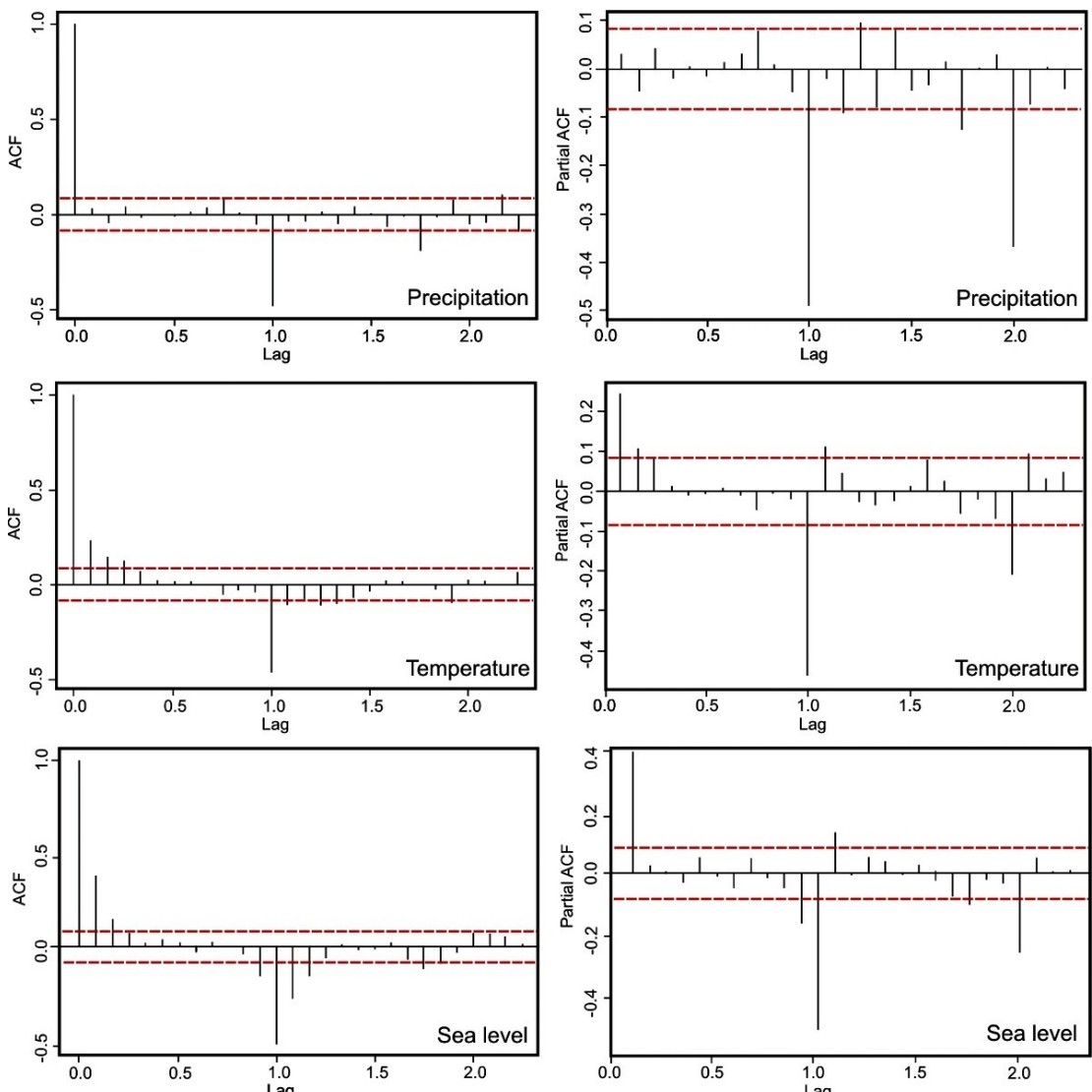

**Figure 4.** The course of the autocorrelation (ACF) and partial autocorrelation (PACF) functions for hydrometeorological conditions in Ustka (monthly intervals from 1966–2015).

Simple stationary models for AR($p$) and MA($q$) can be identified using PACF graphs (for AR) and ACF graphs (for MA). The last significant autocorrelation is the lag amount *(h, lag)*, which can be qualified as value $p$ (for AR) and $q$ (for MA). Unfortunately, in contrast to the MA($q$) and AR($p$) models, the mixed ARMA($p$, $q$) models are difficult to clearly identify using PACF and ACF graphs. A characteristic feature of ACF and PACF for ARMA models is their decay as lag $h$ increases. This decay can be only exponential, or both exponential and dampened-sinusoidal [3]. Therefore, in case of ARMA models from ACF and PACF graphs, it is difficult to clearly identify orders $p$ and $q$ (Figure 4). For this reason, goodness-of-fit criteria like AICC and BIC are generally used to identify orders $p$ and $q$.

### 3.3. Estimation of Parameters and Diagnostic Testing of Models

Seasonal models SARIMA(*p,d,q*)(*P,D,Q*)$_s$ and non-seasonal models ARIMA(*p,d,q*) were chosen for analysis of the time-series for hydrometeorological phenomena. The simplest models with the lowest values for the Akaike Information Criterion (AIC), the Corrected Akaike Information Criterion (AICC), and the Bayesian Information Criterion (BIC) were selected (Table 1). The AIC is one of the most popular criteria for selection of statistical models with varying numbers of parameters. However, it tends to overestimate the orders of models. It is therefore more appropriate to use the adjusted AICC and BIC criteria. The final selection of models was also confirmed by Shapiro–Wilk diagnostic tests. The tests confirmed the normality of residual distribution, as the values for W were greater than the established level of significance ($\alpha = 0.05$) (Table 2). The modelling thus yielded a random residual effect in the form of white noise. The ACF and PACF values obtained for the residuals were close to 0, and statistically insignificant for a confidence level of 0.95 (Figure 5).

**Table 1.** Goodness-of-fit criteria (AIC, AICC, BIC) for ARIMA models.

| Place | Hydromoeteorological Condition | Model ($p,d,q$)($P,D,Q$)$_S$ | AIC | AICC | BIC |
|---|---|---|---|---|---|
| **Świnoujście** | | $(2,0,1)(2,1,0)_{12}$ | 2636.53 | 2636.69 | 2662.41 |
| **Ustka** | Air temperature (mean) | $(2,0,1)(2,1,0)_{12}$ | 2354.47 | 2354.68 | 2384.82 |
| **Hel** | | $(2,0,1)(2,1,0)_{12}$ | 2261.75 | 2261.95 | 2292.10 |
| **Świnoujście** | | $(1,0,0)(1,1,0)_{12}$ | 5528.70 | 5528.77 | 5546.04 |
| **Ustka** | Precipitation (total) | $(0,0,3)(2,1,0)_{12}$ | 5784.25 | 5784.40 | 5810.26 |
| **Hel** | | $(0,0,0)(2,1,0)_{12}$ | 5510.82 | 5510.86 | 5523.83 |
| **Świnoujście** | | $(1,0,0)(1,1,0)_{12}$ | 4500.35 | 4500.42 | 4517.69 |
| **Ustka** | Sea level (mean) | $(1,0,0)(1,1,0)_{12}$ | 4658.98 | 4659.05 | 4676.32 |
| **Hel** | | $(2,0,2)(2,1,0)_{12}$ | 4655.67 | 4655.93 | 4690.35 |

**Table 2.** Verification of normality for the distribution of ARIMA model residuals using the Shapiro–Wilk test.

| Place | Hydrometeorological Condition | Model ($p,d,q$)($P,D,Q$)$_S$ | W | $p$ |
|---|---|---|---|---|
| **Świnoujście** | | $(2,0,1)(2,1,0)_{12}$ | 0.98 | <0.001 |
| **Ustka** | Air temperature (mean) | $(2,0,1)(2,1,0)_{12}$ | 0.99 | <0.001 |
| **Hel** | | $(2,0,1)(2,1,0)_{12}$ | 0.99 | 0.002 |
| **Świnoujście** | | $(1,0,0)(1,1,0)_{12}$ | 0.97 | <0.001 |
| **Ustka** | Precipitation (total) | $(0,0,3)(2,1,0)_{12}$ | 0.96 | <0.001 |
| **Hel** | | $(0,0,0)(2,1,0)_{12}$ | 0.98 | <0.001 |
| **Świnoujście** | | $(1,0,0)(1,1,0)_{12}$ | 0.99 | 0.001 |
| **Ustka** | Sea level (mean) | $(1,0,0)(1,1,0)_{12}$ | 0.99 | 0.011 |
| **Hel** | | $(2,0,2)(2,1,0)_{12}$ | 0.99 | 0.002 |

Consequently, the ACF and PACF diagnostic graphs for residuals of decomposition models, the verification of normality for the distribution of residuals using the Shapiro–Wilk test, and the lowest established values for good-of-fit criteria (especially AICC and BIC) allowed selection of the best ARIMA and SARIMA models for hydrometeorological conditions in the Polish Baltic coastal zone. The following models have the smallest prediction errors: MAE, RMSE, MAPE, MASE, Vϵ, (Table 3).

For average monthly air temperature, the seasonal SARIMA $(2,0,1)(2,1,0)_{12}$ and non-seasonal ARIMA (2,0,1) models were identical for the entire Polish Baltic. Two seasonal (*p*) and two non-seasonal (*P*) autoregressive parameters were found. Average air temperature in a given month is thus influenced by non-seasonal air temperature values from that same month in the previous two cycles (years), as well as two seasonal components and randomness from that month in the analyzed cycle (year).

One non-seasonal parameter of the moving average ($q$) was also found, which indicates that thermal conditions in a given month are influenced by the thermal conditions in that same month the previous year.

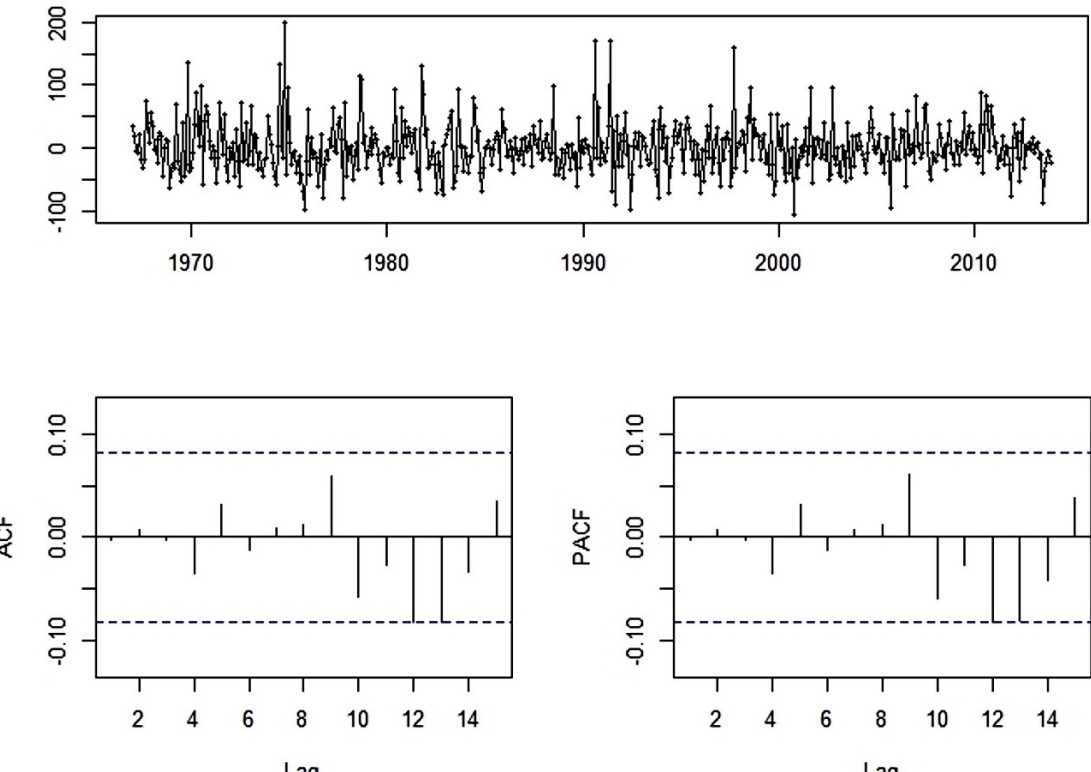

**Figure 5.** The course of autocorrelation (ACF) and partial autocorrelation (PACF) functions for residuals of decomposition models (white noise effect), using atmospheric precipitation in Ustka as an example.

**Table 3.** ARIMA models for hydrometeorological conditions in the Polish Baltic coastal zone and prediction-error criteria (RMSE, MAE, MAPE, MASE, Vϵ).

| Place | Hydrometeorological Condition | Model $(p,d,q)(P,D,Q)_S$ | MAE | RMSE | MAPE | MASE | Vϵ |
|---|---|---|---|---|---|---|---|
| **Świnoujście** | | $(2,0,1)(2,1,0)_{12}$ | 1.92 | 2.58 | 201.07 | 0.35 | 0.8 |
| **Ustka** | Air temperature (mean) | $(2,0,1)(2,1,0)_{12}$ | 1.47 | 1.92 | 205.41 | 0.49 | 1.4 |
| **Hel** | | $(2,0,1)(2,1,0)_{12}$ | 1.35 | 1.77 | 354.11 | 0.48 | 1.5 |
| **Świnoujście** | | $(1,0,0)(1,1,0)_{12}$ | 23.74 | 32.21 | 146.04 | 0.49 | 0.4 |
| **Ustka** | Precipitation (total) | $(0,0,3)(2,1,0)_{12}$ | 29.35 | 40.14 | 111.05 | 0.46 | −0.01 |
| **Hel** | | $(0,0,0)(2,1,0)_{12}$ | 23.85 | 31.70 | 140.03 | 0.47 | 0.4 |
| **Świnoujście** | | $(1,0,0)(1,1,0)_{12}$ | 9.80 | 12.93 | 451.68 | 0.44 | 1.6 |
| **Ustka** | Sea level (mean) | $(1,0,0)(1,1,0)_{12}$ | 11.39 | 14.89 | 185.61 | 0.44 | 1.2 |
| **Hel** | | $(2,0,2)(2,1,0)_{12}$ | 11.20 | 14.71 | 160.70 | 0.42 | 0.4 |

The models for average sea level are characterized by relatively small spatial variability. The parameters of the ARIMA (1,0,0) and SARIMA (1,0,0)(2,1,0)$_{12}$ models are identical for the central coast (Ustka) and west coast (Świnoujście). No MA($q$,$Q$) parameters were found for these areas. Average seal level is influenced only by AR($p$,$P$) parameters. The average sea level in a given month is determined by 1 ARIMA (1,0,0) parameter and 1 SARIMA (1,0,0)(1,1,0)$_{12}$ parameter. On the east coast in Hel (ARIMA (2,0,2) and SARIMA (2,0,2)(2,1,0)$_{12}$), two autoregressive parameters (seasonal and non-seasonal) and two non-seasonal moving-average parameters were found to be significant. Therefore, based on the non-seasonal models, it can be stated that the average monthly sea level in the Gdańsk coastal region

(Hel) is influenced by a previous period one cycle longer ($p$ = 2) than that in the Koszalin coastal region (Ustka) and the Szczecin coastal region (Świnoujście) ($p$ = 1).

Among the hydrometeorological phenomena analyzed, atmospheric precipitation is characterized by the greatest spatial variability. Different modelling parameters were found at each site. The shortest period preceding total precipitation in a given month was in Świnoujście, where only one seasonal and one non-seasonal $(1,0,0)(1,1,0)_{12}$ parameter were found to be influential. The longest period preceding total precipitation in a given month was in Ustka, where three non-seasonal, moving-average parameters and two seasonal autoregression parameters $(0,0,3)(2,1,0)_{12}$ were found to be influential. In Hel, no non-seasonal parameters were found to be influential, and only two seasonal autoregressive parameters $(0,0,0)(2,1,0)_{12}$ were found to be influential.

Model parameters for the time-series were estimated using one-time seasonal differencing ($D$ = 1). The modelling revealed greater spatial homogeneity for air temperature and sea level than for atmospheric precipitation (Table 3).

## 3.4. Model Implementation and Forecasting

Forecasting values for hydrometeorological conditions on the basis of real data is very important for analysis of the time-series in this study. The appropriateness of the models used was confirmed by out-of-sample accuracy analysis. The ARIMA models were very accurate for all measurement stations for the test period. Out-of-sample accuracy analysis of hydrometeorological conditions in Ustka (Figure 6) is a good example of the appropriateness of the models and the forecasts therefrom. The values forecasted were similar to the real values, and the 95% confidence level was rarely exceeded. Most such cases were for average monthly sea level and average monthly air temperature (<10). For total monthly atmospheric precipitation, the 95% confidence level exceeded significantly less often (<5). However, extreme events such as rainfall, causing cliff erosion may occur during a given month. Forecasted values were under- or overestimated in a similar number of cases. This also confirms the appropriateness of the models. In the test period, the greatest accuracy and monthly dynamism of real changes were found for the 4-year period from 2004 to 2007. For this reason, the short-term forecast of the hydrometeorological conditions was generated for the 4-year period from 2016–2019.

This study provides forecasts for the next 48 months from 2016–2019, for prediction intervals with 80% and 95% confidence levels. No significant trends in the increase or decrease of air temperature, atmospheric precipitation, or sea level were found for this forecasted period. However, certain seasonal differences among the forecasted hydrometeorological conditions were visible. For average monthly air temperature, the forecast revealed small, seasonal fluctuations that deviate from the average values from 1966–2015. However, for the winter months, the model showed air temperatures higher by around 1 °C for 2016, and in particular for 2017. In case of atmospheric precipitation and sea level (especially sea level), the models showed a significantly greater and more frequent amplitude of monthly deviations from average values than in case of air temperature. For atmospheric precipitation, the greatest frequency of deviations from average values was found for 2016. Particularly visible is instability of the order of ±30 mm in summer months, where a higher total atmospheric precipitation was forecasted for 2017 than for 2016 and 2019. The forecast for average sea level, however, revealed a high frequency of seasonal deviations of maximally ±10 cm from the average values (Figure 7).

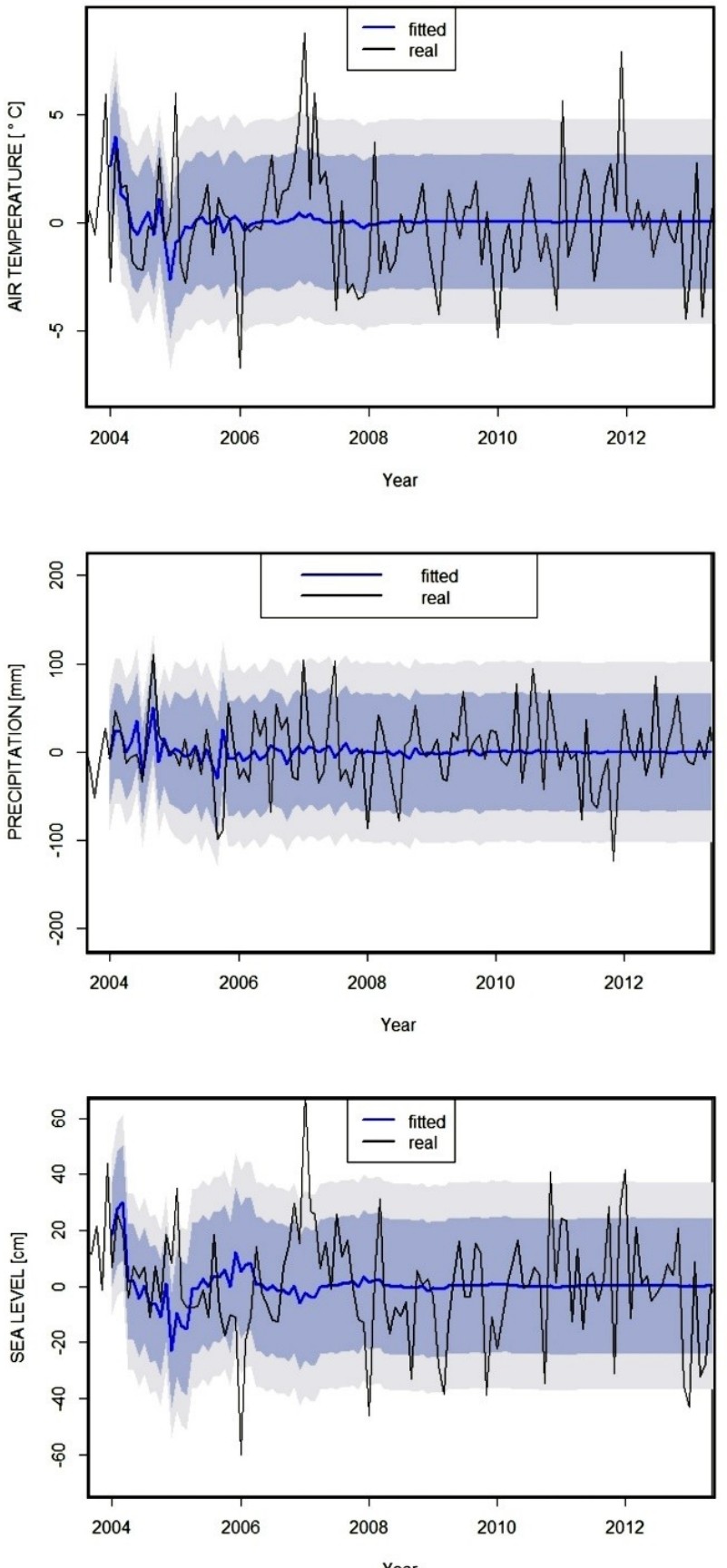

**Figure 6.** Out-of-sample accuracy analysis for hydrometeorological conditions in Ustka.

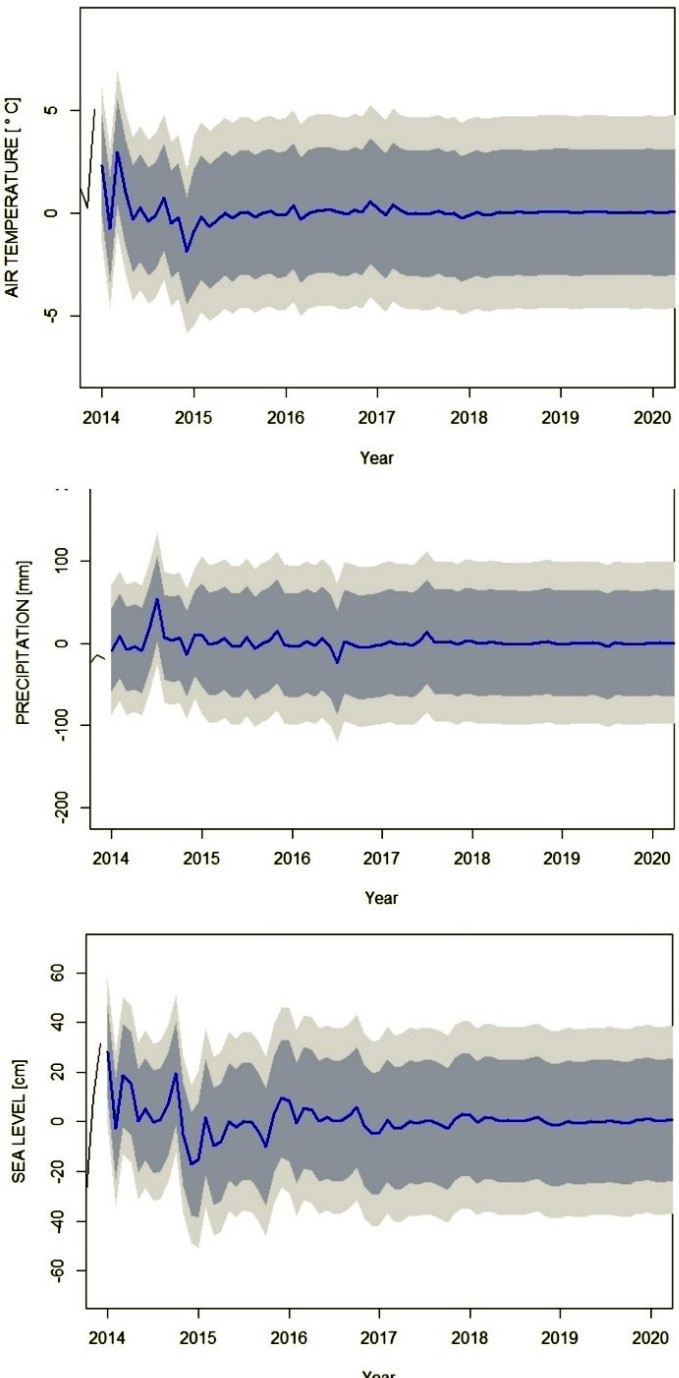

**Figure 7.** Short-term ARIMA model forecasts (2016–2020) for average monthly air temperature, total monthly atmospheric precipitation, and average monthly sea level in Ustka.

## 4. Discussion and Conclusions

Time-series for hydrometeorological conditions in the Polish Baltic coastal zone are best presented by the additive model. Seasonal fluctuation of monthly values for average sea level, average air temperature, and total atmospheric precipitation did not deviate from the trend. Fluctuation amplitude was constant, and the values did not depend on the intensity of the phenomena in time, and were expressed in absolute terms, in the units in which they were measured. The values obtained from the seasonal SARIMA models for the hydrometeorological conditions concern only the Baltic coast. The seasonal variability of the average sea level in the Polish Baltic coastal zone is not identical to

the seasonal variability of open waters. The different seasonality near the coast is partially a result of gradients in atmospheric fluxes at land and sea boundaries, seasonal patterns in coastal circulation, and other topographical elements such as rivers [42]. Sea level data from the coastal zone reveal substantial spatial heterogeneity of sea level regularities (trends and seasonality), reflecting the diverse local and regional processes that impact tide gauge records [43].

The ARIMA models used for the Polish Baltic coastal zone are somewhat spatially homogenous. This is especially true of the models for average monthly air temperature, which are identical across the entire coastal zone $(2,0,1)(2,1,0)_{12}$. Very similar are the models for average monthly sea level across the central and west coast $(1,0,0)(1,1,0)_{12}$. The model for the east coast was determined to be slightly different $(2,0,2)(2,1,0)_{12}$. Monthly values for average air temperature and sea level are thus heavily influenced by regional and continental factors. In contrast to those for air temperature and sea level, the models used for atmospheric precipitation were different for each site. It testifies to large spatial inhomogeneities of precipitation. Among the parameters modelled, autoregressive factors $AR(p)$ were found to have a significantly greater impact on hydrometeorological conditions than moving averages $MA(q)$ with an influence of up to three parameters. An example of these regularities are the monthly models for Ustka: average air temperature $(2,0,1)(2,1,0)_{12}$, atmospheric precipitation $(0,0,3),(2,1,0)_{12}$, and average sea level $(1,0,0)(1,1,0)_{12}$. The ARIMA models made in the R program are able to capture the dynamics of the time-series sensible forecasts.

Hydrometeorological events in the test period (2004–2015) exceeding the 95% confidence level were a result of their high variability and monthly irregularity in individual years. This is especially true of average air temperature and sea level. For example, average air temperature in Ustka in January of 2007 was 5.0 °C, and in 2008 −3.8 °C, where the average for the entire 1966–2015 period was −0.1 °C. Furthermore, total atmospheric precipitation in Ustka in August of 2010 was 193.8 mm, where the average for the entire period was 80.0 mm. However, in November of 2010, total atmospheric precipitation was 3.1 mm, where the average for the entire period was 70.1 mm. The ARIMA model uses monthly rainfall data that generates fewer errors than daily data. Occasionally, exceeding the 95% confidence level may reflect the occurrence of an extreme event during a given month, e.g., precipitation. Extreme rainfall can generate cliff erosion. Extreme precipitation causing landslides on cliffs exceeded the monthly rainfall standard by over 200% [44]. The threshold value is the total sum precipitation exceeding 40 mm and 90 mm within 15 days [44,45].

The average sea level in Ustka in January of 2007 was 552 cm and in January of 2006 was 482 cm, where the average for the entire period was 508 cm. The high irregularity of hydrometeorological conditions in the Polish Baltic coastal zone may be linked to climate change [46] and the increased frequency of extreme events, especially in terms of air temperature and sea level [47]. For this reason, the accuracy of forecasts for certain months may be limited. Nevertheless, such sporadic cases do not diminish the value of the decompositions and forecasts conducted in this study, as the hydrometeorological conditions in the Baltic coastal zone are characterized by significant unpredictability, and short- and long-term variability [48–50]. Although the chosen models cannot predict exact air temperature, atmospheric precipitation, or sea level, they can aid environmental planning and decision-making.

Time decomposition of hydrometeorological conditions based on monthly data did not reveal long-term trends. Yet on the basis of annual values from the last half-century in the Polish Baltic coastal zone, we can observe a statistically significant upward trend in average annual air temperature by ~0.3 °C for every 10 years [51], and an upward trend in sea level by ~3 cm for every 10 years [52]. The actual average increase in sea level was estimated to be 1–2 cm for every 10 years by mareographic measurements [53], and 3–4 cm for every 10 years by satellite measurements [54]. For long, two-hundred-year time-series, the eustatic increase in Baltic sea level is estimated to be 1.3 cm for every 10 years [55,56]. The absence of trends in the decomposition of hydrometeorological conditions conducted for this study also confirms the high variability of monthly values for those conditions from 1966–2015, and thus their reduced predictability.

The ARIMA modelling takes into account the rhythm of the season's sequences in the annual cycle, and multi-year and decade variability, thus, making it possible to forecasting of hydrometeorological conditions forecasts, both in the future (forecast) and in the past (hindcast)—as retrospective modelling. The ARIMA only considers the time variability of a given hydrometeorological element. It does not analyze other variables that determine the value of a given hydrometeorological element. Therefore, ARIMA modelling has some difficulties for dynamic downscaling, operational forecasting (newcast), and forecasting with a scenario [55].

The temporal variability of hydrometeorological parameters (formulated by ARIMA modelling) can refer to the 21st century Regional Climate Models (RCMs). In the South Baltic region, the annual air temperature is expected to increase in the 21st century by 2–3 °C, with an additional increase in atmospheric precipitation by 0–10% in the summer and 10–20% in the winter season [46]. For the southern Baltic coast, according to RCM models, increase in frequent daily rainfall, both in winter and summer is anticipated [57]. It is predicted that in the 21st century, the sea level of the ocean will increase the loss of land ice masses and the thermal expansion of ocean water from 28 to 61 cm [58]. An absolute increase in sea level in the Baltic Sea is estimated at 80% of the global average. For the south and south-west coast of the Baltic Sea, the estimated relative increase in the level would be particularly high, around 50–60 cm [59].

Using the auto.arima function, the designated models were subject to verification in terms of manual differencing and establishment of autoregressive and moving-average parameters. The best of the manual models was almost identical to the automatic model. This was clear from their very similar prediction errors. The ARIMA models presented in this study yielded results comparable to those of exponential smoothing models, and better than those obtained using the seasonal naïve method. The ARIMA models made in the R program, and their verifications with ARIMA and GARCH (Generalized Auto-Regressive Conditional Heteroskedasticity) models in Python, were very similar. Evaluation of prediction accuracy shows that ARIMA models work well when coupled with other eco-environmental models and can be used as a supplemental tool for short-term environmental planning (up to 10 years) and decision-making [60,61]. The ARIMA modelling of hydrometeorological elements in the Baltic Sea zone gives similar results to modelling: Discrete Wavelet Transformation (DWT), Singular Spectrum Analysis (SSA), and Empirical Mode Decomposition (EMD) [26]. The above ARIMA modelling for hydrometeorological conditions can therefore be considered reliable for the Polish Baltic coastal zone.

On the basis of the above analysis, the following general conclusions can be formulated:

- The additive model best presented the time decomposition of hydrometeorological conditions on the Baltic coast.
- The ARIMA models used for the Polish Baltic coastal zone are somewhat spatially homogenous for average monthly air temperature and sea level. Monthly values for these conditions are, thus, heavily influenced by regional and continental factors. However, atmospheric precipitation is characterized by high spatial heterogeneity and significant influence of local factors.
- Among the parameters modelled, autoregressive factors AR(p) were found to have a significantly greater impact on hydrometeorological conditions than moving averages MA(q) with an influence of up to three parameters.
- In the years 1966–2015, hydrometeorological conditions showed a large variation in values. However, time decomposition of hydrometeorological conditions based on monthly data did not reveal long-term trends.
- The forecast of hydrometeorological conditions for the years 2016–2020 did not show significant deviations of monthly values. This did not exclude the occurrence of extreme events, where hydrometeorological values will exceed the range of 95% trust level.
- Climate change causes irregular hydrometeorological conditions and more frequent occurrence of extreme events. Therefore, short-term (several-year) forecasting of hydrometeorological conditions in a short time interval (hourly, daily) is difficult. A lower risk of errors occurs in the case of a monthly interval.

The ARIMA modelling for hydrometeorological conditions is reliable for the research area. It provides important information to help in environmental planning and decision-making.

**Author Contributions:** Conceptualization, J.T. and M.H.; Data curation, J.T. and M.H.; Formal analysis, J.T. and M.H.; Funding acquisition, J.T. and M.H.; Investigation, J.T. and M.H.; Methodology, J.T.; Project administration, J.T. and M.H.; Resources, J.T. and M.H.; Validation, J.T.; Visualization, J.T. and M.H.; Writing—original draft, J.T. and M.H.; Writing—review & editing, M.H.

**Funding:** The APC was funded by the Polish Ministry of Science (Project Supporting Maintenance of Research Potential of the Department of Physical Edu., Health and Tourism at Kazimierz Wielki University no. BS/2016/N1).

**Acknowledgments:** Hydrometeorological data were obtained from the Institute of Meteorology and Water Management, National Research Institute in Warsaw.

**Conflicts of Interest:** The authors declare no conflict of interest.

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
