# Peer review of "Time Decomposition and Short-Term Forecasting of Hydrometeorological Conditions in the South Baltic Coastal Zone of Poland"

_geosciences, doi:10.3390/geosciences9020068_

Round 1
Reviewer 1 Report
34-36 The study is most important for the coast dynamics. But the main factor influencing the abrasion is the wind speed and direction. And the study does not consider these parameters.
It seems that the information presented in the “study area” chapter is the results of authors’ analysis of the data. If yes it is better to put them in the “results chapter”. If no please, provide references.
138-139 ARIMA (p,d,q) modelling is one of the most effective methods of time series forecasting
139 [32-33]. The references are too archaic (1975, 1982). If this model is used in recent studies, please, provide here references.
225-227 The authors assert that “No long-term trends were found…”, however lines 30-31 indicate that “The climate of the Polish Baltic coastal zone is getting warmer, the sea level is rising, and the frequency of extreme hydrometeorological events is increasing”. This is the contradiction. Please indicate that in Abstract you mean that “Time decomposition of hydrometeorological conditions based on monthly data did not reveal long-term trends”
229-230 the results are very trivial and similar to those in lines 112-117 and 121-123
Provide units for parameters at fig. 3 as it is done for fig.6.
It is recommended to show the conclusions more clear (numbered, for example). Now they are combined with the discussion and are indistinct.
Author Response
Thank you for your review.
From a linguistic point of view, the article was translated as a scientific paper by translator from a professional translation agency. This was followed by proofreading by a Native Speaker Jack Ramsey Zagorski.
The following supplementation was made according to the information provided.
The wind speed and its direction determine the intensification of aeolian processes and storm surges. Wind factor belongs to the main determinants of the sea coastal zone functioning. However, the most important hydrometeorological conditions include air temperature, precipitation and sea level. These conditions determine the most rapid geomorphological changes, e.g. mass movements, rinsing and beach, coastal dunes erosion. In addition, there was no homogeneous wind data series for the period considered. It was difficult to calculate reliable monthly values based on wind data.
The information presented in the research area chapter has been moved to the results chapter.
Line 139 has been updated with current references.
Line 225-227 is completed as suggested
Line 229-230 results have been supplemented
Units for parameters in Figure 3 have been supplemented
The conclusions are clearly presented.
Recommended changes are included in the manuscript.
With best regards
Marcin Hojan, Jacek Tylkowski
Reviewer 2 Report
The article is well organized and is logically solid too while needing improvements in the originality perspective. I find this article suitable for publication after the following major issues are resolved. Major Comments: Figure 6 and the corresponding arguments are not solid enough to validate your model. In my view, the good model should reproduce well (1) the variability of the time series (e.g. a box plot of the standard deviation of the 100 realizations of the synthetic time series and how it compares the observed standard deviation); (2) the extreme values. In Line 318, you mentioned that the 95% confidence level was “rarely” exceeded, but I argue that approximately 5 percent of the observed monthly rainfall should exceed this boundary. This is especially important in that the natural disasters are mostly related to the values exceeding this boundary. Would you add discussions on this matter? The beginning (left) part of the time series shown in Figure 6 seems to be unstable and then it stabilizes later on, which contradicts the stationarity assumption of the ARIMA model. Please provide a solid logic for this issue. Minor Comments: Line 62: You may want to refer to some more examples here as follow (not necessary at all, BTW): Yoo, J., Kim, D., Kim, H., & Kim, T. W. (2016). Application of copula functions to construct confidence intervals of bivariate drought frequency curve. Journal of Hydro-environment Research, 11, 113-122. Park, J., Onof, C., & Kim, D. Review of Manuscript hess-2018-267 A Hybrid Stochastic Rainfall Model that Reproduces Rainfall Characteristics at Hourly through Yearly Time Scale.
Author Response
Thank you for your review.
From a linguistic point of view, the article was translated as a scientific paper by translator from a professional translation agency. This was followed by proofreading by a Native Speaker Jack Ramsey Zagorski.
The following supplementation was made according to the information provided.
The ARIMA model was built for monthly data due to smaller prediction errors than for daily data. During the test period (Figure 6), greater variation in value was at the beginning. This may be due to the nearer neighborhood of the learning period and a better reference to real data. In addition, ARIMA modelling has the tendency to somehow flatten of the forecast value, along with moving away from real data. The stability of the time series has been confirmed by diagnostic tests, the course of the autocorrelation (ACF) and partial autocorrelation (PACF) functions , and the course of autocorrelation (ACF) and partial autocorrelation (PACF) functions for residuals of decomposition models (white noise effect) (table 2, fig. 4, 5). The forecast in test period (fig. 6) shows a small number of events exceeding the 95% confidence level. In the test period, there was not a large number of extreme events recorded on a monthly basis. Especially, that monthly average values are taken into account, not the maximum and minimum values. Of course, during this period there were extreme thermal, precipitation and sea level events. But they mainly concerned daily values. Therefore, in the ARIMA modelling (for monthly values) high frequency of extreme events was not recorded. The article is quite extensive, therefore it does not contain some statistical analyzes. Therefore, statistical studies suggested by the reviewer (e.g. a box plot of the standard deviation of the 100 realizations of the synthetic time series and how it compares the observed standard deviation) will be presented in the next article, which will concern the verification of the forecast accuracy (in next year).
Line 318 - supplemented with a discussion of values exceeding the confidence level of 95%.
Line 62 Reference is made to the examples given.
Recommended changes are included in the manuscript.
With best regards
Marcin Hojan, Jacek Tylkowski
This manuscript is a resubmission of an earlier submission. The following is a list of the peer review reports and author responses from that submission.
Round 1
Reviewer 1 Report
General comments:
The article creates a logical whole. As a result of the analytical process, reader obtain a forecast of climate change on the Polish coast of the Baltic Sea for the next few years.
Selected stations (Świnoujście, Ustka, Hel) are representative for the southern Baltic.
The quality of data is guaranteed by the Polish State Institute of Meteorology and Water Management.
A 50-year measuring period is sufficient for this analysis.
The ARIMA model was used correctly. The model was tested and adapted to local Hydrometeorological conditions.
Analysis has shown and confirmed that atmospheric precipitation is characterized by the greatest spatial variability. The sea level and air temperature show a long-term increasing trend.
Suggesting minor changes:
The results are consistent with global climate changes. So I suggest to quote one of the Reports of The Intergovernmental Panel on Climate Change (IPCC) https://www.ipcc.ch/
The sea level data also come directly from the Institute of Meteorology and Water Management in Warsaw danepubliczne.imgw.pl, as written in the text? Or from the Maritime Office?
Reviewer 2 Report
In this paper, the authors have used ARIMA model to examine the hydrometeorological irregularities along the south Polish Baltic coast. The paper is well written and the methodology is well explained along with a good graphical representation of the results obtained from the ARIMA model. However, the paper seems very technical and without much science in it. The authors haven't investigated the scientific causes behind the irregularities in hydrometeorological parameters such as air temperature, sea level, and precipitation. I would recommend publication of this paper after some minor revisions.
1) In Figure 2 I recommend the authors to replace the roman numerical with the name of the months.
2) In the discussion section, I would suggest the authors to include a discussion on changes in the climate parameters and synoptic weather events during the study period. I believe that would greatly enhance the scientific merit of this paper.
Reviewer 3 Report
The paper is very interesting and it is evident great analyses that have been done. Despite this, the progress of description, data and method explanation and the presentation of results are too confused, chaotic, poorly organized, and even the results do not explain what is assumed in the abstract. Bibliography are too poor and not enough recent.
Title: Time decomposition and Short-Term Forecasting of Hydrometeorological Conditions in the South Baltic Coastal Zone of Poland
Authors: Jacek Tylkowski, Marcin Hojan
The paper try to identify behaviour of air temperature, atmospheric precipitation and average sea level in the South Baltic Coastal Zone of Poland.
ARIMA modelling has been used.
The paper is very interesting and it is evident great analyses that have been done. Despite this, the progress of description, data and method explanation and the presentation of results are too confused, chaotic, poorly organized, and even the results do not explain what is assumed in the abstract. Bibliography are too poor and not enough recent.
The introduction does not explain the problem does not speak about the problem on time and on study area, does not deal with scientific evolution and study area features. All bibliographic references are completely missing.
Materials and methods represent Study area, it is necessary to correct the mistake.
Bibliography is too poor.
Methodology does not have enough bibliography and not enough recent
Results are not explain thoroughly.
Discussion and conclusion are too write and it is difficult to follow information, results and evaluation. It would be better to summarize results with tables or graphics.
Reviewer 4 Report
I was not impressed by the manuscript. It is a simple study with a poor introduction and poor Discusion and conclusions also. Large timeseries were examined and some conclusions were drawn, but the presentation of the manuscript is not good and needs improvement.